# Spatial Structure of Lightning and Precipitation Associated with Lightning-Caused Wildfires in the Central to Eastern United States

Brian Vant-Hull [1,*] and William Koshak [2]

1    City College of New York, New York, NY 10031, USA
2    NASA Marshall Space Flight Center, Huntsville, AL 35808, USA; william.koshak@nasa.gov
*    Correspondence: bvanthull@ccny.cuny.edu

**Abstract:** The horizontal storm structure surrounding 92,512 lightning-ignited wildfires is examined in the mid to eastern sections of the United States from 2003 to 2015 using Vaisala's National Lightning Detection Network (NLDN), NCEP's Stage IV gauge-corrected radar precipitation mosaic, and the US Forest Service's Fire Occurrence Database. Though lightning flash density peaks strongly around fire ignitions on the instantaneous 1 km scale, on the hourly 10 km scale, both the lightning and precipitation peaks are typically offset from fire ignitions. Lightning density is higher, and precipitation is lower around ignition points compared to non-ignition points. The average spatial distribution of total lightning flashes around fire ignitions is symmetrical, while that of precipitation and positive flashes is not. Though regression is consistent with the claim that positive flashes have a stronger association with ignition than negative flashes, the statistical significance is ambiguous and is contradicted by an unchanging positive flash fraction in the vicinity of wildfires.

**Keywords:** wildfire; lightning; ignition; dry lightning

## 1. Introduction

In the last two decades, lightning has been responsible for roughly 14% of the reported forest fire ignitions in the United States (Schultz [1] as extracted from the dataset of Short [2]). In contrast to the more common human-caused ignitions, lightning ignitions are nearly equally likely to occur in sparsely inhabited areas where the response time is often slower or nonexistent so that the total acreage burned by lightning ignition outweighs that due to human causes [3,4]. Most such ignitions are attributed to "dry lightning", i.e., lightning flashes in the presence of little or no rain [4–6]. Our previous work [7] demonstrated that for data binned on regional and annual scales, the correlation between dry lightning and wildfire ignitions can be increased by including a regionally adjustable dry period before each candidate flash. This current work examines storm structure associated with lightning ignition at the km and hourly scale, including differences between positive and negative flash patterns.

Previous authors have examined dry lightning primarily in the context of large-scale atmospheric patterns, such as a dry layer below the convectively active layer, finding an increase in ignitions per flash for such conditions [6]. Modeling that accounts for large-scale transport affecting moisture and stability has shown skill in predicting wildfire breakouts [4,5,8]. An alternate way for dry lightning to occur is the horizontal displacement of flashes from the precipitation associated with it, which has rarely been studied and will be examined to some extent in this work.

Flash ignition is also dependent on the current magnitude and duration, and it is commonly assumed that a long continuing current (LCC) is necessary for ignition [9]. Positive flashes are four times more likely to have an LCC than negative flashes [10,11], so should be more likely to cause ignitions, yet the literature shows mixed results. For

example, Anderson [9] modeled wildfire ignitions based on the assumption that only LCC would ignite fires and found correlation coefficients ranging from 0.3 to 0.76, though other complicating variables, such as rainfall, moisture content, and survival time, were modeled rather than observed. Nausler [12] found a significant increase in the ratio of positive to negative strikes in the vicinity of fire ignitions, though cautioned that the greater tendency of positive strikes to land outside the precipitation core could contribute to the relationship. Schultz et al. [1] found that 90% of the lightning flashes closest to ignition points were negative, mirroring the ratio of all negative to positive flashes, and thus, indicating ignition is independent of LCC, agreeing with Flanagan and Wooten [13].

This work examines average horizontal patterns of lightning and precipitation associated with lightning-ignited wildfires compared to those not associated with wildfire ignition. The spatial coincidence of lightning and precipitation peak density relative to individual fire ignitions is part of the pattern analysis. The differences between positive and negative flash patterns and ignition efficiency complete this examination. Our lightning dataset employs radio time-of-arrival and triangulation [14] to single out flashes that strike the ground, matched to a gauge-adjusted radar precipitation product for a gridded estimate of the amount of precipitation that reaches the ground [15]. This work focuses on atmospheric phenomena only rather than the equally crucial fuel condition [16,17], which will be treated as random background noise, an approach that is only feasible when thousands of observations can be averaged together for each point displayed.

To our knowledge, the only other study closely aligned to that presented here is the 2020 work of MacNamara et al. [18], which examined the fine-resolution statistics of lightning and rainfall in the vicinity of lightning-ignited fires. They found that fire ignitions, spatial lightning, and rain peaks were rarely co-located, with lower rain rates and slightly higher flash densities found in the immediate vicinity of ignition points. Positive flashes were no more likely to cause ignitions than negative flashes. Though our work uses a different fire dataset and a coarser precipitation dataset over a different (and larger) time and space domain, it is important to recognize that we independently confirm nearly all their qualitative results and are in reasonable agreement with their numerical results. The main analytical extension in this current work is the comparison of the characteristics of storms that are and are not associated with fire ignitions, while MacNamara et al. compared lighting and rain that is immediately adjacent to ignitions to that which is nearby.

Following a description of the datasets, the temporal and horizontal distributions of lightning around ignition points are examined on the km scale. Precipitation is introduced, and its horizontal distribution relative to lightning is explored on the scale of tens of km. After a statistical assessment of the effects of positive versus negative flashes on lightning ignition, this paper concludes with a comparison of the characteristics of storms that are and are not associated with fire ignitions.

## 2. Data Sets and Methods

### 2.1. Data Sets

This study is built around a fire ignition dataset with associated lightning and rain. The dataset includes CONUS data east of 114 W longitude and from years 2003 to 2015 to avoid irregularities in the precipitation and lighting databases as discussed below.

### 2.1.1. Fire Occurrence Database

The fire database is extracted from the US Forest Service's Fire Program Analysis Fire Occurrence Database (FPA-FOD) produced by the National Fire Incident Reporting System [2]. It includes reporting from federal, state, and local agencies, and as reporting is voluntary, the database is best described as extensive rather than comprehensive. Reporting times are recorded without attempt to estimate the ignition time, which is taken to be typically within the same day as the reporting time. Though the ignition point of fires on federal lands would be estimated via ground survey, the majority of fires in this dataset are east of the Rocky Mountains, thus primarily on private land. The locations of fires in these

cases are given by the street address of the property, unlikely to align closely with the fire locations on large properties. Only those fires reported as caused by lightning are used for this analysis, totaling 92,712. By associating this dataset with lightning strikes (as carried out in this work), Schultz et al. [1] found that 95% of fire ignitions were within ~5 km of the stated location. Roughly half could not be associated with lightning on the same day, attributed to smoldering before breaking out into full fire.

In addition to the attributes used in this study as listed above, the fire database also includes the reporting source, name of fire (if one is given), eventual fire area, and assumed cause, including debris burning, recreation, vehicular, power generation, etc. Lightning ignitions are termed "natural".

### 2.1.2. National Lightning Detection Network

Lightning data are derived from Vaisala's National Lightning Detection Network (NLDN), which consists of a network of about 200 VLF/LF radio frequency receivers over CONUS that combine time-of-arrival and triangulation technologies to locate the location of lightning flashes to an accuracy of 0.5 km and several microseconds [13,19,20]. Flash detection efficiency is estimated at 90–95% [19]. Peak current of the first return stroke, polarity, and multiplicity (i.e., the number of strokes per flash) are available in the dataset. The best accuracy is achieved for flashes, with currents exceeding 15 kA [20,21], but this restriction is only applied for the gridded data described below to provide a continuous current distribution from positive to negative. A significant upgrade to the NLDN occurred in 2002–2003, which motivated us to restrict our analyses to NLDN data from 2003 to later [20]. The data are available from Vaisala as annual files in text format, with one line of records per flash in chronological order.

### 2.1.3. Stage IV Precipitation Radar Data

Precipitation data are taken from NCEP's Stage IV blended radar/gauge gridded mosaic. The hourly accumulated NexRad base scan radar data are calibrated to accumulated gauge data at 4 km resolution by the regional River Forecast centers [15], including manual quality control. From 2004 to 2015, many of the regional centers would not certify results over much of the Rocky Mountains, and so our work is restricted to east of 114 W longitude (Figure 1). The original mosaic is on a 4 km area preserving grid that matches weather models but does not align with the lat/lon grid. For ease of comparison to other data sets in this study, it was rebinned to a 0.1-degree square grid.

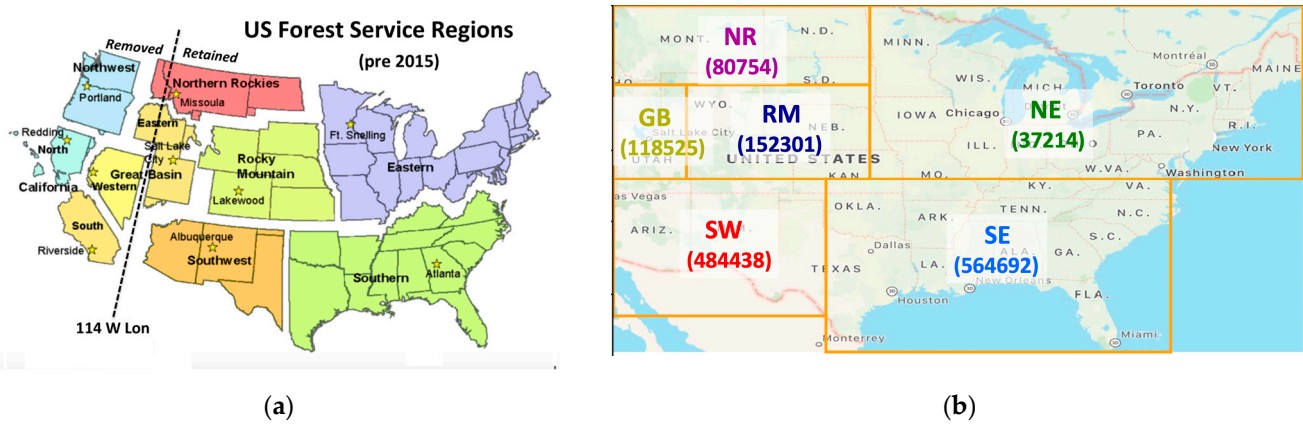

(**a**)  (**b**)

**Figure 1.** National Inter Agency Coordination regions (**a**) with rectangular approximations (**b**). The two-letter initials shown in (**b**) will be used throughout this report. The image (**a**) is adopted from the National Interagency Coordination Center Wildland Fire Summary and Statistics Annual Report, 2003. The total number of lightning ignitions (2002–2015) is listed below for each region.

Due to the inclusion of human quality control, the dataset is passed from the River Centers to NCEP with a lag time of up to 12 h, so may not be available as a monthly bundled dataset until weeks later, making it not useful operationally but ideal for a climate and research dataset, as it extends from 2001 to the present.

*2.2. Methods*

2.2.1. Collating Data to Ignition Events

Each fire ignition was associated with all flashes within a 0.1-degree box centered on the ignition point that fell within $+/-$ 2 days of the reported fire. Distances between flashes and the reported ignition point were calculated, as were area-weighted averages of flash rate and current in radius bins of 1 km up to 5 km. For each flash, the time to a preceding rainfall of 0.1, 0.2, and 0.5 mm/h was tabulated up to a maximum of 7 days (a procedure similar to Vant-Hull et al. [7]).

To include the coarser precipitation data, the procedure pivots to an examination of overall storm pattern in the vicinity of fire ignitions. Fire, lightning, and precipitation data were binned into $5 \times 5$ grids at 0.1-degree resolution centered on each fire ignition point. A separate analysis was performed similar to the above, but for which spatial lightning density maxima were used as grid centers so that fire and non-fire cases could be compared. A lightning peak was classified as "fire" if an ignition occurred at any point in the $5 \times 5$ grid within $+/-$ 1 day.

All the gridded analyses with related figures were performed using the following procedure:

1. A fixed grid of $0.1 \times 0.1$ degrees is imposed on CONUS, with data within each spatial bin accumulated over each hour;
2. Points of interest are detected, defined either by (a) fire ignitions or (b) spatial lightning maxima within this grid system;
3. Grids of $5 \times 5$ are retained centered on each of these points of interest, used to form the spatial average environment around either (a) fire ignitions or (b) lightning maxima.

Since the ignition data is only available at the daily resolution, all occurrences of lightning and rainfall within $+/-$ 24 h of noon on the day of ignition are associated with it. Lightning maxima are classified and averaged separately depending on whether a fire ignition is present within the $5 \times 5$ grid surrounding it. Since fires tend to cluster, this procedure of recentering on each ignition point means that many features will be counted several times. But as we are interested in defining average local environments, this repeat counting is not a distortion.

For the gridded data, lightning is restricted to currents greater than 15 kA and precipitation greater than 0.5 mm/h. In the interest of space, the gridded analysis is not broken down by region.

In many cases, the average pattern is very different from most individual patterns. Since individual cases are of primary interest, statistics of the location of fires and rain peaks relative to spatial lightning peaks are compiled. The data is filtered by Julian day, retaining only those from mid-May to mid-September, reducing the proportion of frontal storms as well as frozen precipitation. The fine-scale analysis is subdivided into regions corresponding to the National Forest Service divisions, as shown in Figure 1. The use of rectangular areas greatly speeds the calculations without a large impact on regional differences. The colors shown are used throughout the analysis to represent various regions, with the entire area denoted in black. The coarse scale analysis is CONUS wide.

2.2.2. Multi Variable Linear Regression

The most common way to demonstrate dependence between variables is linear regression, which requires converting discrete events such as lighting flashes and fire ignitions to continuous variables. This is carried out by averaging counts into area densities. In our case, the $5 \times 5$, 0.1-degree grids described above will contain counts of flashes and fire ignitions relative to each fire ignition, summed over 12 years. The $5 \times 5$ grid averages will

provide 25 paired data points. Using the rule of thumb that there must be at least 10 data points for each predictor variable [22] affords us 2 predictor variables. The natural choice is negative and positive flashes to assess their relative importance to ignition, as shown in the following equation:

$$\text{Ignition density} = k_p \bullet (\text{pos flash density}) + k_n \bullet (\text{negative flash density}) + I_o \qquad (1)$$

With $I_o$, $k_p$, and $k_n$ as coefficients to be determined, ideally, $I_o$ should equal zero.

As described in the analysis section, other variables such as precipitation and compound variables such as (flash density) × (current) were discarded due to the low correlation between the predicted and observed ignition density.

## 3. Results and Analysis

### 3.1. Flash Density versus Days from Ignition

The frequency of lightning flashes as a function of days relative to the stated day of ignition appears in Figure 2. The normalized compilation of all flashes exhibits a very similar peaked pattern among all the regions except the southeast, which is skewed towards flash densities prior to the stated ignition day. Typically, 90% of flashes that occur do so within +/− 1 day of the fire ignition, so the analysis is limited to +/− 2 days (note that for roughly 50% of fires not attributed to human causes, there are no associated flashes in this time range, hence the attribution to smoldering lightning ignitions referred to in the introduction). Storms may occur multiple times during this 5-day period, so to assess the accuracy of the storm ignition data, it is best to look at periods with only one storm.

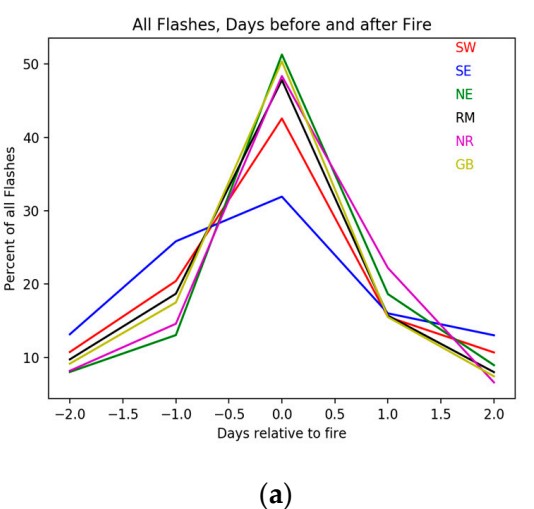
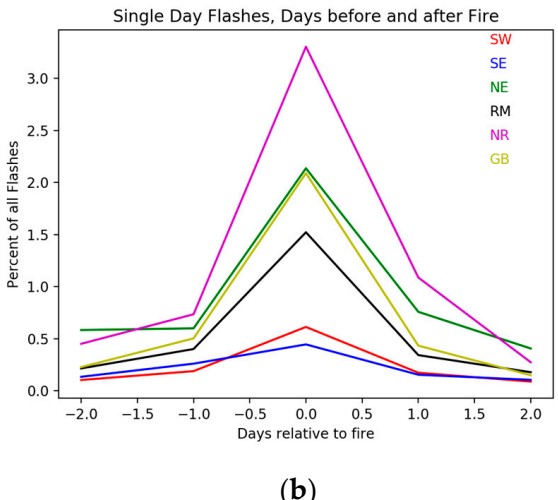

(**a**)          (**b**)

**Figure 2.** Temporal distribution of lightning flashes relative to reported day of fire ignition. (**a**) Distributions for the case of any number of storms during the 5-day period. (**b**) Distributions for the case of only one storm during the 5-day period. (**b**) is weighted by the same total flashes as (**a**).

This is carried out on the right side of Figure 2, still weighted by the total number of flashes. This indicates that the Northern Rockies are most likely to have one storm during a 5-day period, while the South-East region is highly likely to have multiple storms, dropping to the lowest percentage of flashes. All patterns remain peaked, though less so in the southern regions. The SE outlier curve on the left is attributed to the Florida peninsula, host to the highest flash density in CONUS [23]. This makes it less likely to have lone storms in a 5-day period, so the SE fraction is greatly reduced on the right. The magnitudes of other regions behave in a similar manner. On average, there are six flashes on the same day within 0.05 degrees of each fire ignition.

### 3.2. Flash Density versus Distance from Ignition

Flash density and total current are shown as a function of distance on the top row of Figure 3, with regional differences shown in color. The densities are greatest in the vicinity of the fire, falling off immediately (in the first km) to low density with a slow decline with distance, resembling the results of MacNamara et al. [18]. The Great Basin and Northern Rocky regions have the most pronounced peak, consistent with the temporal peaks of Figure 2, as densities in space and time are related.

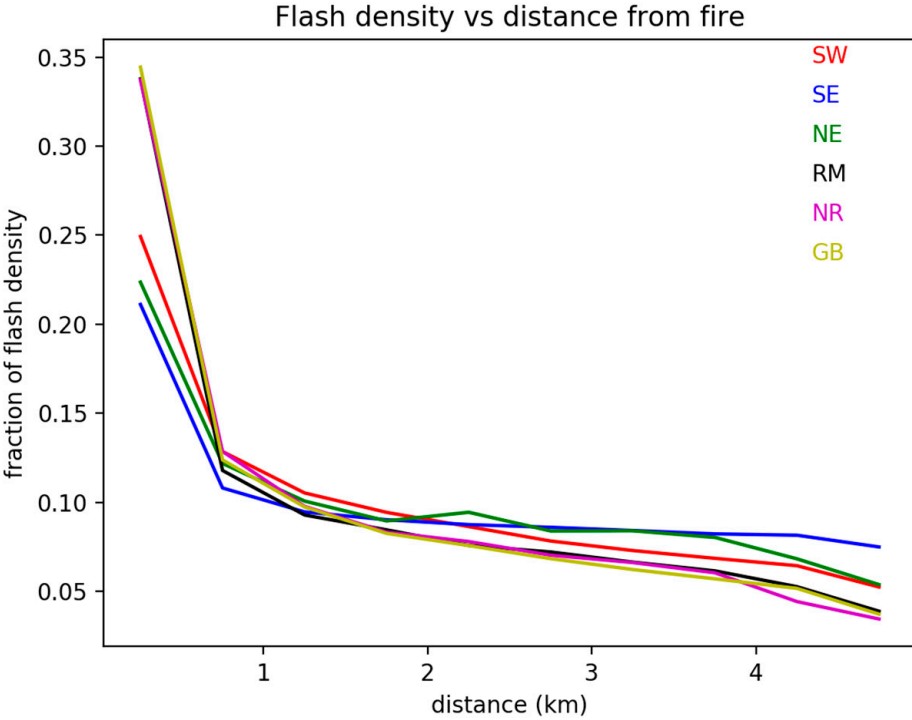

**Figure 3.** Flash density as a function of distance from fire ignitions, normalized by region. Colors correspond to the regions of Figure 1.

### 3.3. Flash Current

Figure 4 examines lightning currents in the vicinity of fire ignitions. The normalized shape of the current histogram (Figure 4a) remains unchanged with the region, i.e., shifted towards the negative current, with a smooth transition into the positive current. If the negative flashes are separated out (Figure 4c), their average current with distance from the fire location decreases slightly with distance [18], and the averages vary up to 20% with the region, with the Southeast currents being the largest. The fraction of the total current that is positive does not show a clear pattern with distance from the fire (Figure 4d), though there are clear regional differences, with the Southeast having the smallest fraction of the positive current. A drop in positive current with distance from a fire ignition (bottom right) is only clear in the Northeast and Southeast. The fraction of positive flashes roughly mirrors the current amplitude of these flashes (Figure 4b). It should be noted that the Southeast region has a much higher negative current and much lower positive current than the other regions, reflecting the same anomalous behavior seen in Figure 2. It should be noted that the division of regions in Figure 1 breaks up an unexplained region of positive lightning in the Great Plains while highlighting the low positive fraction in the Southeast [23].

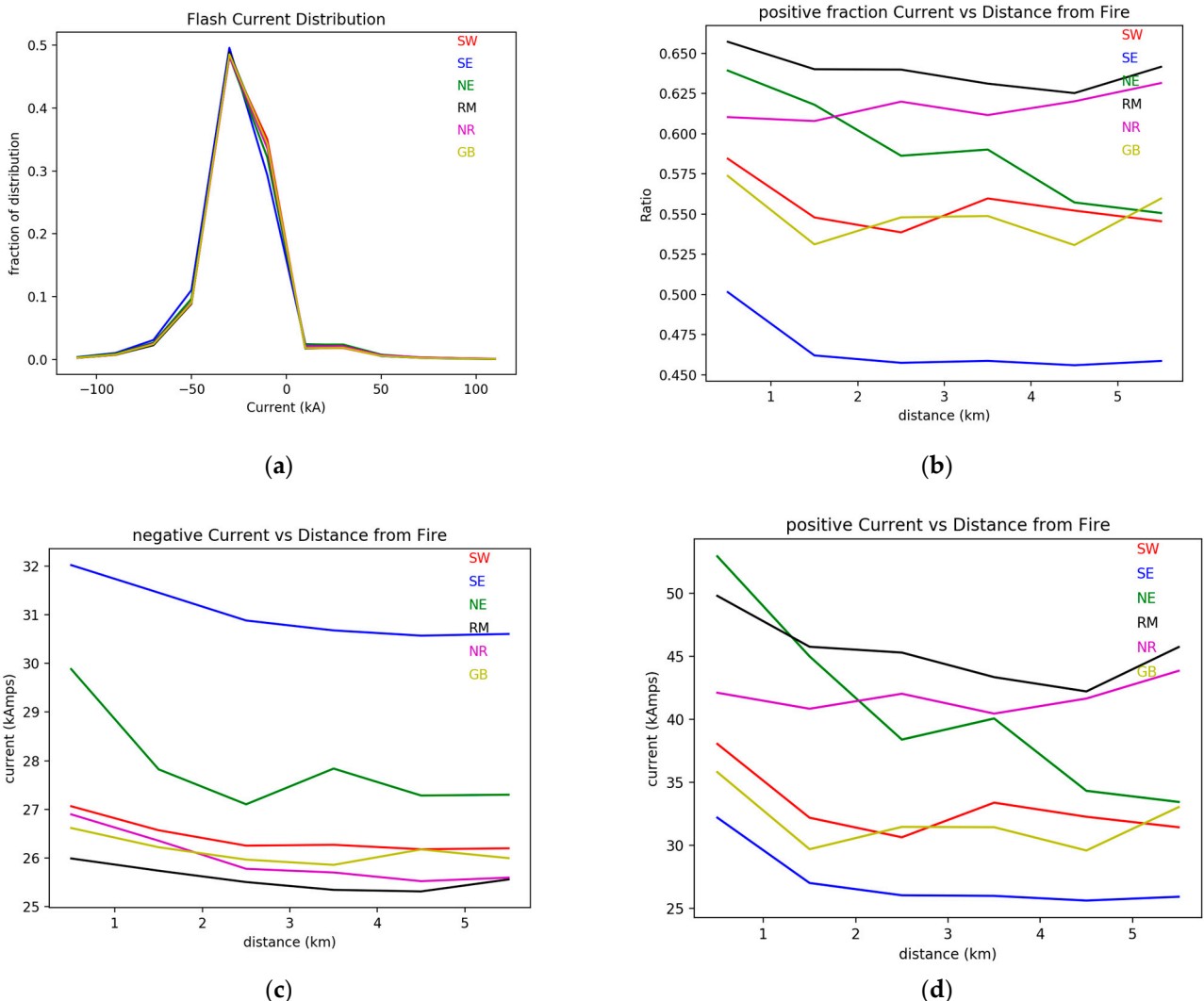

**Figure 4.** Flash current characteristics. (**a**) Normalized distribution of flash currents, by region. (**b**) Fraction of total current that is positive as a function of distance from fire ignition, separated by region. (**c**) Current of negative flashes as a function of distance from fire ignition, separated by regions. (**d**) Current of positive flashes as a function of distance from ignition, separated by region.

### 3.4. Spatial Patterns of Precipitation and Lightning

For comparison to precipitation, the 5 × 5 grids at 0.1-degree resolutions were formed for averages of precipitation, flash count, maximum current (positive and negative), and positive flash fraction. These grid averages appear in Figure 5, showing these values accumulated in the vicinity of each fire ignition (refer to Section 2 for a description of how these are formed). Since the box is roughly half the size of the 0.05 deg radius used above (though flashes are strongly peaked), the number of fires drops to 72,411. The flash rate and current peak at the fire ignition points in the center, while the precipitation peak is offset to the Northeast. Though both flash rate and precipitation tend to peak in the convective core of storms, the precipitation continues for a longer period [24], so the hourly average will become displaced in the prevailing direction of travel. The positive flash count is less strongly peaked than the total flash count because positive flashes are more likely to originate in the anvil rather than the core [25].

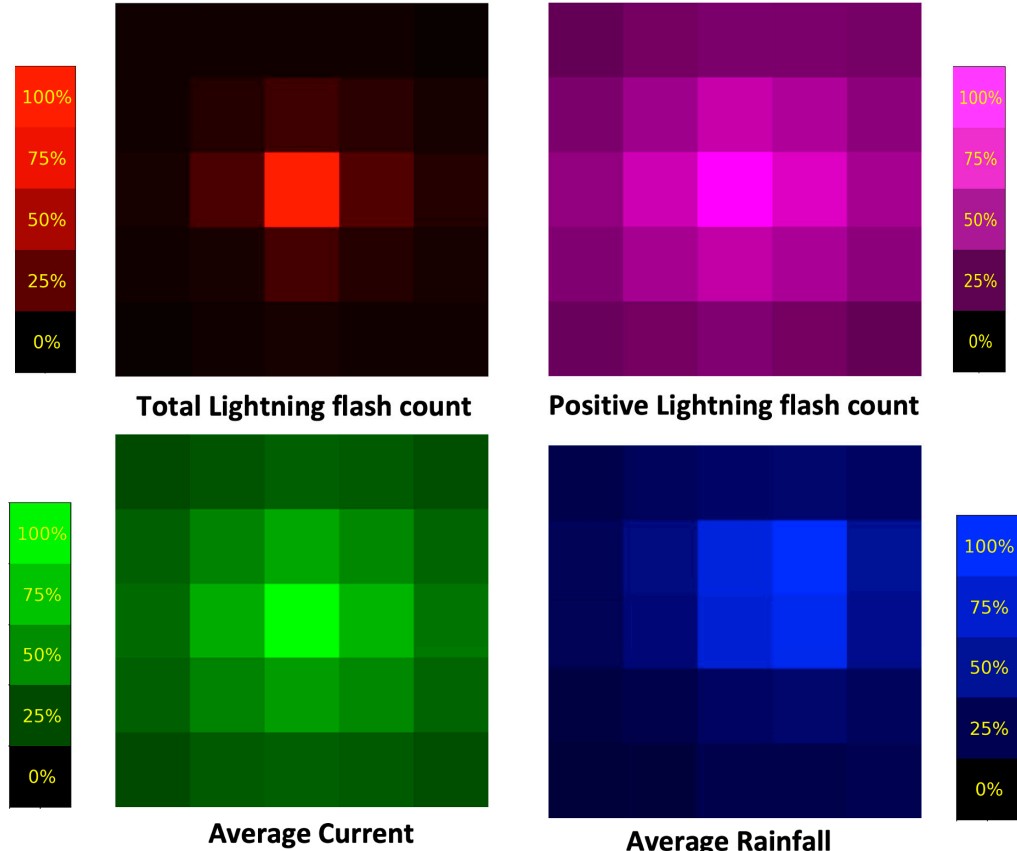

**Figure 5.** Lightning and precipitation in the vicinity of wildfire ignitions. The images represent the averages of values in a 5 × 5 grid, each bin 0.1 degree on a side. The fire ignition is located in the central point. All plots are normalized to their maximum. Statistics are from 72,411 fires.

Positive flashes are rarer than negative flashes but tend to have longer-lasting current flows, thus assumed to be more likely to ignite fires [10,11]. In Figure 6, the ignitions per flash in the vicinity of each ignition are shown. If all ignitions that occurred per each hour period were spaced more than 0.3 degrees apart, the fire density on the left of Figure 6 would be a single spike at the center, as would be the ignitions per positive or negative flashes. Given that fires and flashes cluster around an ignition point, it is rather surprising that the ignitions per negative flash actually are lower at the point of ignition, while the ignition ratio for positive flashes is higher at the center than expected. There are several possible reasons for this. Our approach examines each fire in turn so that the total ignition probability for the central fire is a binary variable always equal to 1, while the flash rate is strongly peaked at the ignition location. This explains the drop in ignition ratio in total lightning as an artifact of the selection process. But to explain why a similar drop is not seen in the positive flash count will require a deeper examination of the storm structure, as carried out in the next section.

All the patterns in Figures 5 and 6 exhibit a peaked pattern (or valley in the case of ignitions per positive flash). To examine the significance of these peaks, a Student T test was applied to see if the inner 9 grid values were statistically distinct from the outer ring of 16 values. In all cases, the difference was significant at the three-sigma level or above.

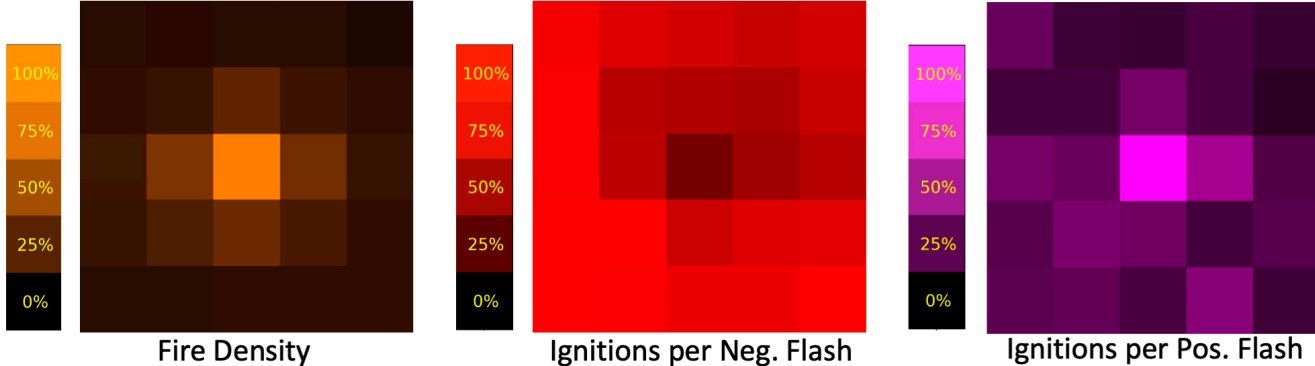

**Figure 6.** Comparing ignitions per flash for negative (**middle**) or positive (**right**) polarity. These are averages centered on each wildfire ignition. For context, the average ignition density is presented (**left**). All diagrams are normalized to their maximum.

### 3.5. Lack in Spatial Coincidence of Ignitions, Lightning, and Precipitation Peaks

The strong central tendency of *total* flash density should not be confused with the density of peak locations during *individual* events. When peak locations alone are examined, the density of lightning peaks is slightly peaked at ~8% above the average at the fire location (graphics not shown due to low contrast), even though the total flash density is strongly peaked over the fire location (Figure 5). This means that for a majority of events, the lightning peak does not coincide with the point of ignition (compared to MacNamara et al. [18]). An analogy would be overlapping Gaussian distributions for which the central peak of the total distribution does not correspond to individual peaks. The rain and lightning peaks are strongly coincident, despite the total rainfall being shifted weakly to the NE (Figure 5).

Individual ignition events can be examined for the occurrence of local peaks in lightning flash density and precipitation, as well as the presence or absence of precipitation or lightning. These statistics appear in Table 1. We see immediately that the most common scenario is no rain or lightning, and the possibility of not having a lightning flash coincident within the same day of ignition is 72.4%. Schultz et al. [1] observed a 50–60% lack of coincidence of lightning within a 5 km radius, which they attributed to smoldering before the fire became evident enough to be reported. A possible reason for the discrepancy is the restriction to data largely east of the Rockies in this current work which means lower reporting rates of small fires in the less inhabited mountains are discounted. It may also be the exclusion of flashes with currents below 15 kA in our dataset.

**Table 1.** Occurrence of precipitation, lightning flashes, and their peaks in the vicinity of reported wildfire ignitions. Statistics based on 92,512 fires.

| % | No Rain | Rain Present | Rain Peak | Row Totals |
|---|---|---|---|---|
| **No Lightning** | 61.1 | 10.0 | 1.3 | *72.4* |
| **Lightning Present** | 3.4 | 14.6 | 1.7 | *19.7* |
| **Lightning Peak** | 1.4 | 5.4 | 1.1 | *7.9* |
| **Column totals** | *65.9* | *30.0* | *4.1* | 100% |

If the row of Table 1 relating to the no lightning cases is ignored, we see that it is roughly 2.5 times more likely for ignitions to occur if they are *not* coincident with a local peak in lightning, perhaps because the rainfall is typically higher at the lightning peaks. This helps explain the drop in ignition efficiency for total flashes near the ignition points; in fact, it is five times more likely for fires to be ignited when offset from the rain peak. But the positive lightning flash density is less strongly peaked. Due to the tripole charge structure of storms, positive flashes are more likely to occur in the anvils and stratus shield

than negative flashes, which concentrate more in the precipitation core [25–27]. The ratio of negative charges to positive is, therefore, much larger in the core, and so even though the ignitions per flash are suppressed due to rain, the number of ignitions is higher. The result is a false sense that the positive flashes are more efficient in producing ignitions in the center, when instead, the increase in negative flashes is responsible for the increase in ignitions.

### 3.6. Regression of Flash Density versus Ignition Density

The other reason why the positive ignition ratio may more closely match the ignition pattern compared to negative flashes is the longer continuing current, which makes positive flashes more efficient at ignition and, therefore, more closely correlated to ignitions. This can be shown via multivariable regression relating the positive and negative flash densities in Figure 6 to ignition density (Figure 7). The plot of all ignitions vs. all flashes provides a strong correlation but has a non-zero intercept.

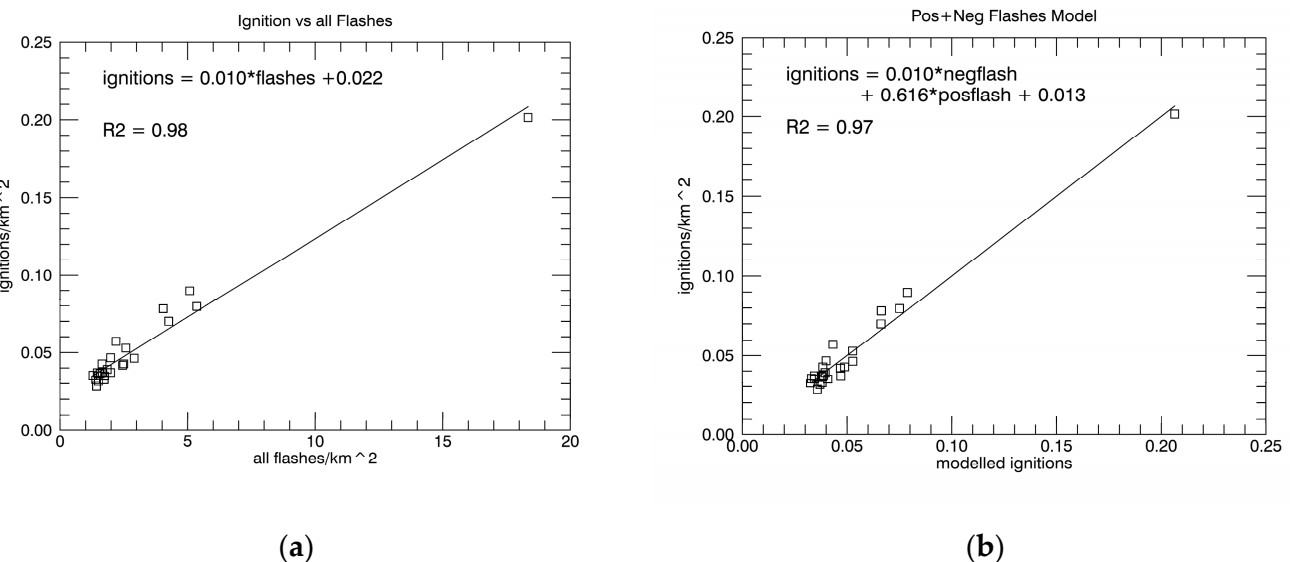

(**a**)  (**b**)

**Figure 7.** Plots of ignition density vs. flash density summed over each $5 \times 5$ 0.1-degree grid centered on each ignition Left: plot of ignitions vs. ALL flashes. Right: Plot of observed ignitions vs. predicted ignitions from multivariable linear regression model based on positive and negative flash density. The R-squared values are adjusted to account for the number of predictor variables.

When the flashes are split into positive and negative, it is seen that each positive flash is roughly 60 times more likely to ignite a fire than a negative flash, and the intercept is nearly cut in half at the sacrifice of a slightly lower correlation. The sum of absolute deviations is also about 5% smaller when positive flashes are included in the model, tipping the balance towards the combined model. These positive indicators, combined with the drop in adjusted correlation mean model improvement due to the introduction of polarity, remains ambiguous and must be cross-checked with other indicators as described below.

It should be emphasized that this linear fit to 25 grid points applies to the results of summing over a large number of cases in which ignition occurred in the immediate vicinity, preconditioning regional fuel state while averaging out local variability. If the fitting was performed to individual flashes, the effects of vegetation would be more apparent, and the correlations would be much lower. The fit would be improved if the point at the top right was slightly higher in the observations. This point corresponds to the central ignition point at which average rainfall increases, which dampens ignition efficiency and is not accounted for in the model. Though removal drops the overall ignition rate ratio of positive/negative flashes down to roughly 50:1, the change in the quality of fit when the polarity is introduced remains as ambiguous as before and thus is not reported.

The more physically reasonable model of relating ignition density to the combined (flash density) × (average current) was slightly less successful, perhaps because ignitions are more likely caused by extreme flashes, and the current averaged over 10 × 10 km squares does not well represent the distribution of individual flashes. A model that included rainfall was also less successful, presumably for the same built-in assumption of gridpoint-wide uniformity of rainfall.

As a check on whether the 60-fold increase in ignition efficiency implied by the fit of Figure 7b is meaningful or not, we can apply the following simple test: is the fraction of fires ignited by positive flashes larger (by a factor of 60) than the fraction of positive flashes overall? We find that about 5% of flashes within 0.05 degrees (roughly 5 km) of a fire ignition are positive. Koshak et al. [23], using the same NLDN dataset, found a national average that varied from 4% to 8% depending on the year, with much of the variability driven by an, as yet, unexplained peak in positive lightning in the northern Great Plains. Similar results are reported in MacNamara et al. [18]. The fraction of fires ignited by positive lightning thus matches the fraction of positive lightning overall, contradicting the claim that positive lightning flashes are more efficient at ignition. From this, we conclude that the linear regression, as applied to area averages, is not an appropriate tool for evaluating the effects of positive lightning, perhaps because the fraction of positive lightning is so small that the added variable is simply fitting noise rather than contributing signal.

*3.7. Comparison of Lightning Density Peaks: Fire versus No Fire*

All the analysis has looked at so far is the conditional statistics in the presence of wildfire ignition. It would be illuminating to compare ignition events to non-ignition events, but then we require an independent definition of an event. A good candidate is the location of flash density peaks so that the properties of lightning peaks with and without co-located fire ignitions can be compared. A total of 4,024,259 lightning peaks without fires were compared to 37,325 peaks associated with fires. A spatial comparison appears in Figure 8, for which each row of fire versus no fire is normalized to the maximum value of that row. A lightning peak was classified as "no fire" if there were no ignitions in the 5 × 5 grid centered on the peak.

Several of the behaviors in Figure 8 are expected; the average flash density is higher, and the precipitation is lower in the presence of fire ignitions. Though the trend is slight, the precipitation does reach a peak coincident with the flash peak if there is no fire, but this shallow peak is translated to the NE if a fire is present. The average positive flash rate and average current are also higher during ignition events, though the weighting of positive flashes to the south while precipitation is weighted to the north (as expected due to storm motion) remains unexplained. The roughly 50% increase in the peak value of positive flash counts from the no-fire to the fire cases matches that of the overall lightning count, though the positive lightning counts exhibit less contrast from the rim to the center peak. From this comparison, there is no indication that positive flashes are more associated with fire ignitions than negative flashes, as the positive flash fraction is essentially unchanged in the peaks.

The same test of peak significance applied to Figures 5 and 6 (end of Section 3.4) was applied to Figure 8; all peaks shown are statistically distinct at the three-sigma level or greater.

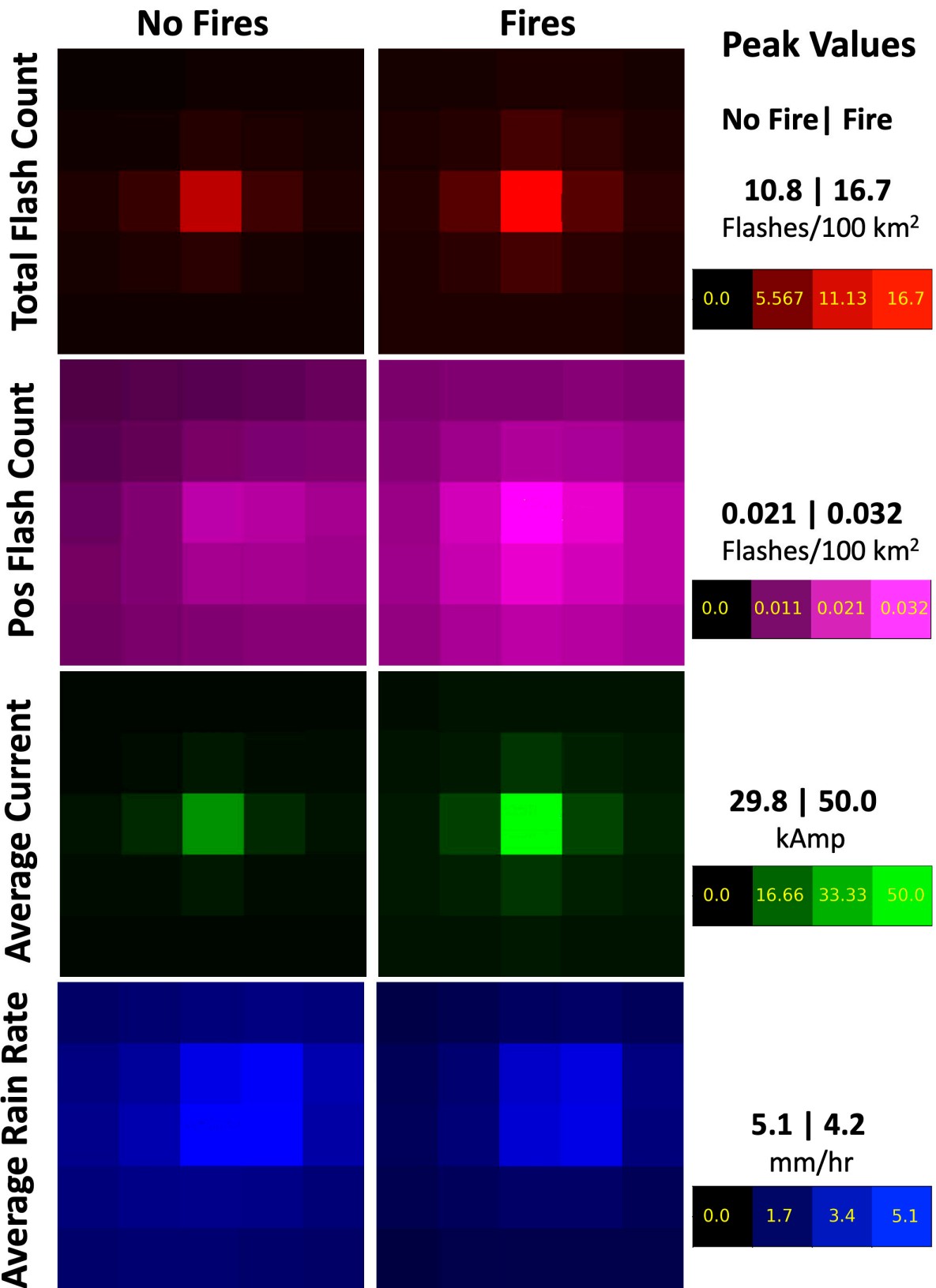

**Figure 8.** Spatial patterns centered on lightning peaks not associated with ignitions (**left**) versus peaks with fires (**right**). Color normalization is relative to the highest value of each row. Images are based on 3,024,259 lightning peaks without fires and 37,325 lightning peaks with fires.

## 4. Discussion: Applicability and Future Work

The most unique aspect of this work is the demonstration that fire ignitions are more likely to occur when the rainfall peak is spatially offset from the lightning peak. Can this be applied to fire forecasting systems such as the widely used Wildland Fire Assessment System (WFAS) [28–31] that estimates wildfire occurrence based on weather predictions and fuel moisture? Spatial offsets between precipitation and lightning could occur because surface rainfall is affected by the horizontal wind field more than lightning. Upper-level wind forecasts are readily available, though, in most cases, lightning would be striking moistened fuel. Alternatively, differences in the timing of peak occurrence combined with storm motion would also result in the horizontal displacement of lightning and precipitation. Since predictions need to include detailed storm modeling, this second mechanism is less applicable to large-scale forecasting. Both these mechanisms would be most relevant in the absence of a dry layer below the storm clouds that evaporates precipitation before hitting the ground [6].

The location of ignitions is relative to lightning peaks, and the storm motion vector would help establish if lightning striking outside the storm would remain dry due to the storm path. This aspect may be important, though difficult to predict, and will be included in the next stage of work.

The National Lightning Detection Network used in this work detects ground strikes but cannot capture continuing currents. The GOES 16 Lightning Mapper (GLM) has complementary capabilities, so a merge of the two instruments would be a large step forward in lightning ignition studies. A continuation of this work comparing long continuing current ground strikes to shorter current strikes will be pursued.

## 5. Conclusions

On a fine scale, lightning flash density drops quickly with distance from an ignition, by a factor of between 2 and 3 within the first km distance before flattening out. Current and positive flash fractions drop more gently with distance, exhibiting a much greater variation with the region that does the total flash count. We saw that roughly 70% of reported fires could not be associated with lightning on the same day, an effect Schultz et al. [1] attributed to smoldering with a similar miss rate of 50–60%.

Moving to the coarser scale 0.1-degree grids, we see that in the presence of fire ignitions, total lightning is more sharply peaked than either positive flash count or current, while the average rainfall is offset to the NE. The ratio of ignitions per flash for total lightning is actually lower at the fire ignition location due to strongly peaked lightning density. For positive flashes, the ignitions per flash ratio are higher at ignition points. However, individual cases show a different pattern from the average patterns. It is unlikely for fire ignitions, lightning peaks, and precipitation peaks in all cases to coincide, being just one of several possible configurations. This agrees with recent results by other researchers [18].

Multivariable linear regression suggests that positive flashes are roughly 60 times more efficient at igniting fires than negative flashes. The positive flash density tends to peak more strongly during ignition events. Though this result is consistent with the assumption that positive flashes are more efficient at fire ignition, we can not claim the case to be statistically definitive, and it is contradicted by the positive flash fractions seen in this work and other recent studies [1,18] that remain unchanged in the vicinity of fire ignitions.

**Author Contributions:** Conceptualization, B.V.-H. and W.K.; methodology, B.V.-H.; software, B.V.-H.; validation, B.V.-H.; formal analysis, B.V.-H.; investigation, B.V.-H.; resources, W.K.; data curation, B.V.-H.; writing—original draft preparation, B.V.-H.; writing—review and editing, W.K.; visualization, B.V.-H.; supervision, W.K.; project administration, W.K.; funding acquisition, W.K. All authors have read and agreed to the published version of the manuscript.

**Funding:** This work was supported by the Precipitation and Lightning Work Package for the Internal Science Funding Model (ISFM) project *Lightning as an Indicator of Climate* under NASA Headquarters (Dr. Jack Kaye and Dr. Lucia Tsaoussi), which supports NASA's participation in the National Climate Assessment (NCA).

**Institutional Review Board Statement:** Not Applicable.

**Informed Consent Statement:** Not Applicable.

**Data Availability Statement:** The Wildfire Occurrence Database is freely available from https://www.fs.usda.gov/rds/archive/Product/RDS-2013-0009.5 (accessed on 29 June 2023). Stage IV gridded and gauge corrected precipitation radar data is freely available from http://data.eol.ucar.edu/dataset/21.093 (accessed on 29 June 2023). The Vaisala Lightning Detection Network archived data is available by request or real-time data available for purchase from https://www.vaisala.com/en/products/national-lightning-detection-network-nldn (accessed on 29 June 2023).

**Acknowledgments:** The authors gratefully acknowledge Vaisala Inc. for providing the NLDN data used in this study as part of the Marshall Mentored Project (MMP, ID #02). We also acknowledge the Stage 4 radar data provided by NCAR/EOL under the sponsorship of the National Science Foundation. The authors gratefully acknowledge the guidance of Karen Short of the U.S. Forest Service in the use of the Wildfire Occurrence Database.

**Conflicts of Interest:** The authors declare no conflict of interest.

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
