# Peer review of "Spatial Structure of Lightning and Precipitation Associated with Lightning-Caused Wildfires in the Central to Eastern United States"

_fire, doi:10.3390/fire6070262_

Round 1

Reviewer 1 Report

The manuscript has correct form, but it is not well-numerated. There are two subchapters 3.2. and also two subchapters 3.5. The authors should correct numeration and check the whole manuscript as well.

The Introduction, the Data Sets and the Results are correct. The analysis is also well.

However, the reference list is too short having in mind the high standards of the journal. Some references are with DOI numbers, but most are without them. The authors should use the Authors Guide.

The authors should take into consideration the results of new research showing the connection between the activity of the Sun (i.e. the solar wind) and forest fires. These papers could help them understand the problems they were dealing with.

Author Response

Reviewer 1   YesCan be improvedMust be improvedNot applicable Is the content succinctly described and contextualized with respect to previous and present theoretical background and empirical research (if applicable) on the topic? (x) ( ) ( ) ( ) Are all the cited references relevant to the research? (x) ( ) ( ) ( ) Are the research design, questions, hypotheses and methods clearly stated? (x) ( ) ( ) ( ) Are the arguments and discussion of findings coherent, balanced and compelling? ( ) (x) ( ) ( ) For empirical research, are the results clearly presented? (x) ( ) ( ) ( ) Is the article adequately referenced? ( ) (x) ( ) ( ) Are the conclusions thoroughly supported by the results presented in the article or referenced in secondary literature? (x) ( ) ( ) ( )   The manuscript has correct form, but it is not well-numerated. There are two subchapters 3.2. and also two subchapters 3.5. The authors should correct numeration and check the whole manuscript as well.   > Numerations have been revised, with extra sections added pursuant to other reviewer's comments.   The Introduction, the Data Sets and the Results are correct. The analysis is also well.   However, the reference list is too short having in mind the high standards of the journal. Some references are with DOI numbers, but most are without them. The authors should use the Authors Guide.   > DOI and official links have been added when available. Given the somewhat unique approach of this work it would seem that too many more references would be padding rather than useful.   The authors should take into consideration the results of new research showing the connection between the activity of the Sun (i.e. the solar wind) and forest fires. These papers could help them understand the problems they were dealing with.   > Looking into this, all papers found on this topic share a common author, which suggests the topic of solar wind effects on fires is not a trend in the literature.  Examining the paper most closely related to this one, "Modelling Of Forest Fires Time Evolution In The Usa On The Basis Of Long Term Variations And Dynamics Of The Temperature Of The Solar Wind Protons", the authors did not find any signal while using linear regression between fire occurrence and solar wind, so switched to a neural network. The data set had 2,654 data points, and was fit with 729 "fuzzy rules" which may be nodes or may be weighting factors, it's not clear.  But this would represent a ratio of data/(fitting parameters) of less than 4 to 1 at best (much worse if each "fuzzy rule" has more than one parameter), which is dangerously close to overfitting and definitely runs the risk of "fitting the noise". It is accepted protocol when using the black box of a neural network to separate into training and validation sets, and if this was done it's likely the signs of overfitting would be evident; but this fundamental step was ignored.  It's rather surprising the reviewers let this pass, and so we can not reference this as representative science.  In the same vein the current work has been edited to clarify our own regression has stayed within the rule of thumb of at least 10 (data points)/(fitting parameter), and a reference added.

Reviewer 2 Report

Summary Statement

This study takes the NLDN lightning data and investigates composites with fire ignitions, precipitation, and polarity of lightning. The study is performed over the CONUS east of the Rockies and finds that lightning density is highest over ignition grid boxes while precipitation is maximum to the NE of the ignition points. Positive lightning flashes are more likely to lead to ignition.

The square shaded box plots present an interesting perspective on this topic which was new to me. Overall, the paper is written clearly and the results are understandable. I think the manuscript would benefit from a tighter structure and edits to the graphics to make it easier to read. I recommend accept with major revision.

Major comments

1.     Abstract: Include years of the study in the abstract. After intro revisions, the abstract could be made stronger.

2.     Introduction: I found the organization difficult in the introduction because it did not make it clear to me what the paper was going to focus on. Dry lightning? Positive/negative flashes? There were bits of information but it did not hold together in a coherent story. Identify the knowledge gap you want to address and organize the introduction text around that. Some revision to the text should address this comment.

3.     Section 2- Data: I wanted to know more ‘factoids’ about the fire occurrence database. How many records and which fields are available. This journal is for a global audience so this information is good to document, and it also makes the results more accessible and convincing. I have similar comments about the NLDN. How many data records and what fields are available? What are the error bars in the data? This is not public data so many people are not exploring it, so the more you can inform, the better to advance knowledge. Also, it would be good to include any evaluations of the NCEP Stage IV gridded rain data. How well does it do in your study regions? What are its strengths and weaknesses?

4.     Section 2 – Methods. I suggest moving 2.4 into another subsection of Section 2 that focuses on methods. All statistical methods used should be described in the methods. The multivariate linear regression model  (mentioned in conclusions) should be described under methods and should include the variables that are important predictors. But include it only if it is important. If it is a teaser for future work, then it should be worded differently, like ‘preliminary work’ or such.

5.     Figures 5, 6, & 8: These figures all need a color bar legend of some sort. How much lower are the darker values than the brighter ones? Also, how many data points (what was sample size) went into each color box plot? In figure 8 the small white font is very hard to read and should be removed. Some type of error bars or uncertainty measures from these plots would strengthen the presentation.

6.     Conclusions: State more clearly what your new findings are. This should match what you define as your knowledge gap in the introduction. The connection between these two parts connect feels confusing right now.

Minor comments

1.     Line 169, the first section is 3.2 under the results section.

2.     Title: I would not use the word variability here but that could be my mindset as I think of time variability. This is the spatial variability of the lightning and its associated factors. You may want to think about just deleting ‘Variability of’ from the title. You could replace this phrase with ‘Spatial structure of’

Author Response

Reviewer 2   YesCan be improvedMust be improvedNot applicable Is the content succinctly described and contextualized with respect to previous and present theoretical background and empirical research (if applicable) on the topic? ( ) (x) ( ) ( ) Are all the cited references relevant to the research? (x) ( ) ( ) ( ) Are the research design, questions, hypotheses and methods clearly stated? ( ) (x) ( ) ( ) Are the arguments and discussion of findings coherent, balanced and compelling? ( ) (x) ( ) ( ) For empirical research, are the results clearly presented? ( ) (x) ( ) ( ) Is the article adequately referenced? (x) ( ) ( ) ( ) Are the conclusions thoroughly supported by the results presented in the article or referenced in secondary literature? ( ) (x) ( ) ( )   Summary Statement   This study takes the NLDN lightning data and investigates composites with fire ignitions, precipitation, and polarity of lightning. The study is performed over the CONUS east of the Rockies and finds that lightning density is highest over ignition grid boxes while precipitation is maximum to the NE of the ignition points. Positive lightning flashes are more likely to lead to ignition.   The square shaded box plots present an interesting perspective on this topic which was new to me. Overall, the paper is written clearly and the results are understandable. I think the manuscript would benefit from a tighter structure and edits to the graphics to make it easier to read. I recommend accept with major revision.     Major comments   1.     Abstract: Include years of the study in the abstract. After intro revisions, the abstract could be made stronger.   > years of study has been added, but with a limit of 150 words not much more can be included.   2.     Introduction: I found the organization difficult in the introduction because it did not make it clear to me what the paper was going to focus on. Dry lightning? Positive/negative flashes? There were bits of information but it did not hold together in a coherent story. Identify the knowledge gap you want to address and organize the introduction text around that. Some revision to the text should address this comment.   > There has been some revision and additional phrases added.  One reorganization is rather nonstandard but seems to help the flow: there is only one paper we found that matches the topic of the current work. It was originally grouped with the other research, but when put after the introduction of the focus of the paper it helped bring out the uniqueness of these two studies.  Though in a sense this reorganization "buries the lead" it seems to clarify things somewhat.   3.     Section 2- Data: I wanted to know more ‘factoids’ about the fire occurrence database. How many records and which fields are available. This journal is for a global audience so this information is good to document, and it also makes the results more accessible and convincing. I have similar comments about the NLDN. How many data records and what fields are available? What are the error bars in the data? This is not public data so many people are not exploring it, so the more you can inform, the better to advance knowledge. Also, it would be good to include any evaluations of the NCEP Stage IV gridded rain data. How well does it do in your study regions? What are its strengths and weaknesses?   > Some additional facts about the data sets have been added, but a full evaluation of how well they compare to other data sets would comprise a study in itself.  Where such studies are available they have been referenced.   4.     Section 2 – Methods. I suggest moving 2.4 into another subsection of Section 2 that focuses on methods. All statistical methods used should be described in the methods. The multivariate linear regression model  (mentioned in conclusions) should be described under methods and should include the variables that are important predictors. But include it only if it is important. If it is a teaser for future work, then it should be worded differently, like ‘preliminary work’ or such.   > The multi variable regression is now described in its own section under methods.  It is included mainly as caution to others that the results of regression must align with other results in order to be meaningful, and the topic of ignition efficiency of positive lightning has a lot of murky results in the literature.  Extra discussion of this point has been added.   5.     Figures 5, 6, & 8: These figures all need a color bar legend of some sort. How much lower are the darker values than the brighter ones? Also, how many data points (what was sample size) went into each color box plot? In figure 8 the small white font is very hard to read and should be removed. Some type of error bars or uncertainty measures from these plots would strengthen the presentation.   > Colorbars have been added and the number of points are included both in text and captions. > Uncertainties will be different for counts such as lightning strikes and continuous variables such as precipitation.  For counts in a cell there is no distribution and no uncertainty estimate unless some other division such as variability by year is done.  For a continous variable such as precipitation the standard deviation would need to be divided by root(N) to be meaningful, and with counts in the thousands we will be dividing a stanard deviation on the order of the average by a factor of roughly 30 to 100; only a few percent uncertainty at most.  But precipitation should likely be done in lognormal space.  Current is more complex because of the assymetrical positive/negative distribution, so cannot be done in the same lognormal space as precipitation.  All this is a long way to say be careful what you ask for, it is very messy.  It would be better for such work to be relegated to an appendix, and we will consult with the editors about this.   We did do a Student T test in earlier versions of the work to see if the peaks in positive lightning were significant at better than 1% confidence, but removed it once a shift in analysis produced over 1000 data points per grid box and the confidence level exceeded 0.1%.  Though not the same as a grid point level error bar, it is a clear indication that the main conclusions are significant, and we have included a note to this effect for figures 5,6 and 8.   6.     Conclusions: State more clearly what your new findings are. This should match what you define as your knowledge gap in the introduction. The connection between these two parts connect feels confusing right now.   > The conclusions felt fairly clear to us as written, but perhaps the "newness" is unclear because our results largely mirror a recent similar paper.  Some small rewording has been done.   Minor comments   1.     Line 169, the first section is 3.2 under the results section.   > repaired along with the extra sections added.   2.     Title: I would not use the word variability here but that could be my mindset as I think of time variability. This is the spatial variability of the lightning and its associated factors. You may want to think about just deleting ‘Variability of’ from the title. You could replace this phrase with ‘Spatial structure of’   > Good suggestion, applied.

Round 2

Reviewer 2 Report

I have read through the revised paper and the author’s revisions have
addressed my concerns. The paper is more accessible and clear now.
I recommend that the paper is ready for publication.

I had one comment
1. Line 252, re Figure 4. This is more of a curiosity. Do you know of
any explanation on why the SE US has a much lower number of positive
lightning flashes than other CONUS regions? If there are thoughts on why
this the case, then it would be good to add a sentence around 252. I
wondered if it was because the SE is humid. But that is not consistent
with the next lowest being the GB region. The SE just stands out.

Author Response

Unfortunately there is no more understanding of why the SE has low positive fraction than why the midwest has high positive fraction!  See the maps in reference 23.  I feel the anomalous high positive fraction in the midwest is muted by the division of regions, while the SE is highlighted.  A sentence has been added to point this out.